

# Towards a component-based system model to improve the quality of highly configurable systems

Tehseen Abbasi[1], Yaser Hafeez[1], Sohail Asghar[2], Shariq Hussain[3], Shunkun Yang[4] and Sadia Ali[1]

[1] University Institute of Information Technology, PMAS Arid Agriculture University, Rawalpindi, Punjab, Pakistan
[2] Department of Computer Science, COMSATS University Islamabad, Islamabad, Pakistan
[3] Department of Software Engineering, Foundation University Islamabad, Islamabad, Pakistan
[4] School of Reliability and Systems Engineering, Beihang University, Beijing, China

## ABSTRACT

Due to ever-evolving software developments processes, companies are motivated to develop desired quality products quickly and effectively. Industries are now focusing on the delivery of configurable systems to provide several services to a wide range of customers by making different configurations in a single largest system. Nowadays, component-based systems are highly demanded due to their capability of reusability and restructuring of existing components to develop new systems. Moreover, product line engineering is the major branch of the component-based system for developing a series of systems. Software product line engineering (SPLE) provides the ability to design several software modifications according to customer needs in a cost-effective manner. Researchers are trying to tailor the software product line (SPL) process that integrates agile development technologies to overcome the issues faced during the execution of the SPL process such as delay in product delivery, restriction to requirements change, and exhaustive initial planning. The selection of suitable components, the need for documentation, and tracing back the user requirements in the agile-integrated product line (APL) models still need to improve. Furthermore, configurable systems demand the selected features to be the least dependent. In this paper, a hybrid APL model, quality enhanced application product line engineering (QeAPLE) is proposed that provides support for highly configurable systems (HCS) by evaluating the dependency of features before making the final selection. It also has a documentation and requirement traceability function to ensure that the product meets the desired quality. Two-fold assessments are undertaken to validate the suggested model, with the proposed model being deployed on an active project. After that, we evaluated the proposed model performance and effectiveness using after implementing it in a real-world environment and compared the results with an existing method using statistical analysis. The results of the experimental study proofs that the proposed model is practically and statistically significant as compared to the existing method in terms of effectiveness and participants' performance. Hence, the statistical results of the comparative analysis show that the proposed model improved ease of understanding and adaptability, required effort, high-quality achievement, and version management are significant *i.e.*, more the 50% as compared to the exiting method *i.e.*, less than 50%. The proposed model offers to assist in the development of a highly configurable system that achieves the needed quality. Therefore, the proposed model manages the

Corresponding author
Shunkun Yang, ysk@buaa.edu.cn

variation identification, versions control, components dependency for correct selection of components, and validation activities from domain engineering to application engineering.

## INTRODUCTION

Software development is a complex activity that involves knowledge management, fast product development, a competitive market, multiple industrial aspects, and quick advancement in technologies (*Clarke et al., 2016*; *Giray, 2021*). As a means of dealing with all these complexities, using resources efficiently, and establishing control, software development organizations mostly select those methods that help in the execution of the software product development process within a given time. There are many methods available for software development which includes traditional software development life cycles like the waterfall method. The main problem with these methods is that they are not flexible to changes and required more time for documentation and initial planning. This significantly disturbs the time-to-market and may result failing of the software product. On the other hand, agile ensures shorter releases, faster functionality delivery and feedback, timely delivery, and increases quality (*Dove, Schindel & Hartney, 2017*; *Camacho et al., 2021*). The development that would be carried with agile improves the pace of adaptability and development, which is most important to satisfy market demands (*Klünder et al., 2019*).

Software product line (SPL) engineering supports reusable common software resources by following a predefined architecture and plan. The reuse of different predefined features enables product tailoring to make it fit for customer needs (*Aggarwal & Mani, 2019*; *Camacho et al., 2021*; *Al-Hawari, Najadat & Shatnawi, 2021*). SPL becomes a vital paradigm for companies as it favors usability, cost, productivity, quality, and time (*Krueger & Clements, 2017*, *2019*; *Chacón-Luna et al., 2019*; *Bolander & Clements, 2021*). Variability is the capacity of the product framework to be changed, re-configured, expanded, and arranged for use in a particular context, hence becoming a central concern for researchers and practitioners (*Krueger & Clements, 2018*; *Carvalho et al., 2019*; *Wu et al., 2021*; *Ali et al., 2021a*). SPL aims to develop a time-efficient and cost-effective methodology for the HCS by reusing its assets (*Dintzner, van Deursen & Pinzger, 2018*; *Carvalho et al., 2019*; *Ter Beek et al., 2020*). Usually, standalone products adopt the whole variability model, yet most of the features are different (*Abal et al., 2018*).

Nowadays, agile software development and SPL have become more popular in the software development industry and both approaches are an authentic way of software development (*Hohl et al., 2016*; *Hayashi & Aoyama, 2018*; *Aggarwal & Mani, 2019*; *Oriol et al., 2020*; *Kasauli et al., 2021*; *Kiani et al., 2021*). The agile manifesto provides a

better architecture to SPL with integrated methods along with SPLE (*Chacón-Luna et al., 2019*; *Klünder et al., 2019*; *Kiani et al., 2021*). Recently, many researchers tried to investigate both paradigms (*Haidar, Kolp & Wautelet, 2017*; *Hayashi & Aoyama, 2018*; *Krueger & Clements, 2018*; *Klünder et al., 2019*) because both approaches share some common goals like customer satisfaction, limiting costs, reduced time to market, quality, and improved software productivity (*Hanssen & Fægri, 2008*; *Aggarwal & Mani, 2019*; *Klünder et al., 2019*). After combining both methods, the researchers named them agile product line engineering (APLE) (*Hohl et al., 2018*). APLE, the hybrid process model having mutual benefits, satisfies the customers with their common objectives and needs. Moreover, SPL handles variability identification, variability management, and selection of the features. On the other hand, agile just need requirements to deliver the required product (*Mohan, Ramesh & Sugumaran, 2010*; *Abal et al., 2018*; *Chacón-Luna et al., 2019*; *Kiani et al., 2021*). These approaches are correspondingly categorized as reactive and proactive software engineering approaches. Hence, both approaches have the same objective of improving software development efficiency.

The main issues are dynamic variation and configuration which causes irrelevant selection of components and variability management for reuse and restructuring due to lack of documentation and component repository management during HCS development based on APLE. Therefore, the objective of this research is to address the issues identified from the existing literature and described in this section like the adaption of automatic documentation of the initial document and the code. Moreover, the selection of the components or features to reduce the dependency between the features and ensure the quality of the final product variant by using test-driven development and requirement tracing functionality, and finally the configuration of both processes to be suitable for HCS development.

## Research contributions

To overcome the mentioned problems, we develop and present a hybrid process model preserving the benefits of both *i.e.*, Agile-SPL and HCS. Following are the contributions of this paper:

- A significant review of literature has been carried out to understand the existing studies about agile SPLE and HCS. The review described that there is a need for a component-based system model consisting of SPL-based features and developed under agile methodology to improve the quality of HCS during verification identification, version control, and management for reuse of components during development.
- To improve quality and productivity of HCS for SPL based component-based system a QeAPLE Model is proposed for APLE for HCS based SPL to manage variabilities and relevant selection of components depending on user feedback and reusability for identification, managing, and selection of variation and their relevant components for reuse and version control.
- To automate the QeAPLE model developed a prototype based on the designed algorithm for the correct and relevant selection of components for reuse to manage variability

during the development of SPL-based HCS products. The implemented in a real-world environment to evaluate the performance of prototype and practice theory into practice.

- To evaluate the effectiveness of the proposed model, an empirical study is performed by the practitioners with the help of the prototype in the real scenario for a practical implication of the QeAPLE model.

- After that performed a comparative analysis in an empirical study to evaluate the effectiveness of the QeAPLE model in terms of commonalities and variabilities management in HCS with the existing method. We also evaluated the performance of participants using the QeAPLE model as compared to existing methods. The existing model which we used for comparative analysis selected from literature *i.e.*, Arkendi model (*Mollahoseini Ardakani, Hashemi & Razzazi, 2018*).

- The QeAPLE model provide guidelines and directions for researchers and industrialists during dynamic variability management and selection of components for reuse and restructuring in APLE during HCS development.

The rest of the paper is structured as follows: "Related Work" discusses the literature review. Furthermore, it also discusses the research gap identified in the existing work. "Quality Ensured Agile Product Line Engineering Process Model" provides the details about the proposed process model and its components. It also describes the functioning of the proposed model and its post and preconditions. "Experimental Evaluation" describes the evaluation of the proposed model and a comparison of the experimental results with the existing method. Finally, "Conclusion and Future Work" concludes the research work and provides the possible future directions.

## RELATED WORK

There are several research studies found in the literature that tends to integrate agile software development with product line engineering to gain the benefits of both processes. In *Hohl et al. (2016)* led a subjective study about integrating the agile process with SPLs which is helpful for organizations to incorporate the end-user changes rapidly and launch the software to the market in a timely fashion. Furthermore, they distinguish that the advancement procedure can be improved by transparency, cooperation, adaptability, productivity inside the developers' group, and software quality grounded by the reuse within the profit range. The highly configurable system requires the integration of the features that are least dependent upon each other and could be modular as high as much (*Meinicke et al., 2016*; *Abal et al., 2018*; *Ter Beek et al., 2020*). The agile SPL model should be capable of providing the product with such characteristics. The quality of HCS is difficult to analyze because of multiple variations of a single product. Consequently, a comprehensive testing mechanism is required for the achievement of product quality (*Parejo et al., 2016*; *Abal et al., 2018*; *Kasauli et al., 2021*). *Yoder (2002)* provided a tailoring approach to manage the new variant according to the product line variant, and then integration, as well as delivery of the final variant is carried out using an agile development process. The main limitation in this approach is that the documentation part and the component selection parts are not clearly described. It also does not address the HCS.

Similarly, in *Ghanam & Maurer (2010)*, the researchers tend to alter the variation integration mechanism using the code refactoring method. The main problem with the proposed method was that the mechanism is not optimized for the selection of the independent features. *Carbon et al. (2008)* in their work improved the integration by test-driven development (TDD) addition. This ensures the quality of the new variant. However, it does not provide the mechanism to check component dependency, and development of the configuration system at the time of feature selection. The researchers in existing literature provide a comprehensive solution for the adoption of the integrated APLE model (*Haidar, Kolp & Wautelet, 2017*; *Hayashi, Aoyama & Kobata, 2017*; *Hohl et al., 2018*; *Kiani et al., 2021*). *Haidar, Kolp & Wautelet (2017)*, proposed a comprehensive model for the agile product line engineering process, still, it does not support feature selection or components to make software highly modular. The proposed model not only provides test-driven development for quality assurance but also provides insights into documentation and variation management. The key issues in this approach are the negligence of feature selection before using them in TDD, and incompatibility with HCS. To solve these issues, a comprehensive method is required which will not only select the least dependent component but also deliver the automatic documentation along with requirement analysis for better variation management. The focus of this research is the execution of comprehensive steps required to use agile techniques in iterative. The approach used is reactive, which considers both application engineering AE and domain engineering DE. The main limitation of this research work is that it doesn't talk about the quality of the end product. Moreover, it doesn't discuss highly configurable systems support in the proposed approach. Similarly, other works discussed above (*Yoder, 2002*; *Carbon et al., 2008*; *Ghanam & Maurer, 2010*), have the same common issues in their contributions. *Hohl et al. (2018)* and *Kiani et al. (2021)* identified that the application engineering process doesn't provide detailed feedback to the domain engineering phase, which is mainly responsible for version management. The researcher improved the APLE process by making it semi-automatic and allowing the application engineering process to send feedback to the domain engineering process. The main limitation of the existing approach is that it cannot improve end-product quality. check the feature dependency while selecting the features for the new product. To improve the APLE model, a scoping mechanism for the APLE process is proposed (*da Silva, 2012*). It allows improved version management and provides better version control. The main limitation of this study is that the process is not favorable for HCS, as HCS requires an improved feature selection process by first checking their dependency. Moreover, it does not talk about achieving the quality of the end-product to get the most useful information from the application engineering process, to aid the domain engineering process (*Tian, 2014*). It has been determined that domain engineering requires much information to improve version control. The main drawback of this mechanism is that it doesn't ensure the quality of the end product. Moreover, it doesn't talk about feature dependency checks while selecting features for the new product. *O'Leary et al. (2012)* mainly focused on the application engineering part of APLE rather than domain engineering. The main aim of the proposed mechanism was to ensure product quality. The mechanism tends to improve testing of the

product to ensure the quality of the end product. The proposed mechanism's main limitation is that it does not talk about version control, and the feature selection process is also faulty that needs much improvement. *Cardoso et al. (2012)* identified the need for the APLE model to produce a security surveillance system. To address the problem, the researcher proposed the APLE model for security surveillance system production. The main limitation of this research work is that it doesn't properly focus on the application engineering process and tends to achieve quality by test-driven development. Moreover, the feature dependency is also needed to be analyzed while configuring them to make a new product. Similarly, *Abal et al. (2018)* proposed the APLE framework for large production units and industries. The researcher identified that the existing APLE models are only configured for small and medium enterprises. It needed to be re-tailored for large industries. The proposed framework doesn't support the quality achievement of the product and it doesn't identify the feature dependency while making their selection for a new product variant. In another work, *Hohl et al. (2018)*, performed an analysis for the proposition of the APLE model for the automobile variants. Researchers analyzed that the application engineering process for automobiles is very important compared to the domain engineering process. To provide a comprehensive APLE model, the researcher first identified the appropriate recommendations, and then based on these recommendations, they proposed a novel model for the automobile industry. The main problem with the proposed mechanism is that the mechanism doesn't support quality assurance and variability management. Moreover, the feature dependency check was also missing in the proposed mechanism.

Improvement of version management is also an important aspect. *Ghanam & Maurer (2010)* mainly focused on the improvement of version management for the APLE process. The main improvement they introduced was the refactoring process that provides the classified information for each of the versions. The main drawbacks include the quality check of the product being ignored while the feature dependency is also neglected while selecting the components for a new product variant. Besides version management, improvement in the APLE process to make it fast in the initial planning is also desired. The possible improvement in the APLE process identified in different studies and improves the initial planning of the product. Along with that, the quality checking of the work is also done and highlighted that the proposed mechanism is not able to provide comprehensive version management and feature dependency check.

Apart from providing the APLE model in the automotive industry and surveillance camera production units, literature identified the need for the APLE process for enterprise systems that is relatively complex to handle. Researchers in this research proposed an APLE model for enterprise industries (*Dove, Schindel & Hartney, 2017*; *Hohl et al., 2017*; *Klünder, Hohl & Schneider, 2018*; *Uysal & Mergen, 2021*). The main limitation of this research work is that it doesn't check the feature dependency while selecting new products. In *Hayashi, Aoyama & Kobata (2017)*, *Klünder, Hohl & Schneider (2018)* and *Kiani et al. (2021)* integrated APLE process. This process is typically comprised of the scrum as an iterative application engineering process. The main limitation of the proposed approach is that it doesn't provide much feedback and nor is there any automatic documentation

module. Furthermore, there is a high need to maintain version control, which depends on the feedback that came from the application engineering process. *da Silva et al. (2014)*, *Klünder et al. (2019)*, *Kasauli et al. (2021)* and *Camacho et al. (2021)* emphasized that there is currently no APLE model that completely provides all the details of the integrated process. To address the identified problem, researchers proposed a new, fully comprehensive APLE model with all necessary steps required to produce a new variant iteratively. Still there is limitation of lack of a dependency check while selecting the features. The requirement of a transformation model for converting the production from a traditional SPLE process to an agile SPLE process is significant. *Klünder, Hohl & Schneider (2018)* proposed a new transformation model that helps the industry to follow the APLE model for the production of new variants. The main limitation of the proposed approach is that there was no definition of version control and quality achievement module. Furthermore, the feature dependency check is also a must, which is missing in the proposed approach.

These features are very useful, and hence they are more user centric. The model is built on a merge algorithm to make the feature model more comprehensive and efficient. The main limitation of the proposed approach is that the model does not provide the quality achievement of the final product. Moreover, the proposed model also fails to provide feature dependency and analysis checks before their integration into the new product. *da Silva et al. (2014)* presented a new agility-based approach for scoping the product line details. These details are gathered using communication and interviews with the customer and more focus on user involvement to help the developer to deliver the product of the required quality. The main limitation of the proposed approach is that it doesn't talk about the quality of the product. Moreover, feature dependency is also not checked while selecting the features for the new product variant.

The key issues in this approach are the negligence of feature selection before using them for validation after variation are irrelevant, and incompatible with HCS. Therefore, variation identification, variation management, and mapping are important to manage version control and relevant selection reuse components with proper documentation, repository management, and valid identification of test cases of selected reuse components. To solve these issues, a comprehensive method is required which will not only select the least dependent component but also deliver the automatic documentation along with requirement analysis for better variation management. The summary of a literature review is discussed in Table 1. Therefore, in proposed model resolves the identified problems by correct variation identification, accurate dependency of selected components, and validation of reuse components for variation in a new product.

# QUALITY ENSURED AGILE PRODUCT LINE ENGINEERING PROCESS MODEL

A novel agile-enabled software product line engineering model is introduced based on the scrum process presented in *Mollahoseini Ardakani, Hashemi & Razzazi (2018)*, and the frameworks proposed in *Mellado, Fernández-Medina & Piattini (2010)*. This model will provide support for the configuration and development of highly configurable systems.

**Table 1 Summary of literature review.**

| References | [16] | [17] | [18] | [19] | [20] | [21] | [22] | [23] | [24] | [25] | [26] | [27] | [28] |
|---|---|---|---|---|---|---|---|---|---|---|---|---|---|
| Documentation | ■■ | ■■ | □□ | ■□ | ■■ | ■■ | ■■ | ■■ | ■■ | ■■ | ■■ | ■■ | ■■ |
| Variation management | ■□ | □□ | ■□ | ■□ | ■■ | ■□ | ■□ | ■□ | □□ | ■■ | ■■ | ■■ | ■■ |
| Domain knowledge | ■□ | ■□ | ■■ | ■■ | ■■ | □□ | ■□ | ■■ | ■■ | ■□ | ■□ | □□ | ■■ |
| Commonalities | ■□ | ■□ | ■■ | ■□ | ■□ | ■□ | ■□ | ■□ | ■■ | ■□ | ■□ | ■□ | □□ |
| Version control | ■□ | ■□ | ■■ | ■□ | ■□ | ■□ | □□ | ■□ | ■□ | ■■ | ■□ | ■□ | ■□ |
| Work synchronization | □□ | □□ | □□ | ■□ | ■■ | ■□ | □□ | ■■ | ■■ | ■■ | ■■ | ■□ | ■□ |
| Lack of knowledge reusability | □□ | □□ | ■■ | ■■ | ■■ | □□ | ■□ | ■□ | ■□ | ■■ | ■■ | ■□ | ■■ |
| Configuration Management (HCS) | ■■ | ■□ | ■□ | ■□ | ■■ | ■■ | □□ | ■■ | ■■ | ■■ | ■■ | ■□ | ■□ |
| Component selection | ■□ | ■■ | □□ | □□ | ■■ | ■□ | ■■ | ■■ | ■□ | ■■ | ■■ | ■■ | ■□ |
| Component testing | □□ | ■□ | ■■ | □□ | ■■ | □□ | □□ | ■□ | □□ | ■■ | ■■ | ■□ | ■■ |
| Task allocation for teams | ■■ | □□ | ■□ | ■□ | ■■ | ■■ | ■■ | □□ | □□ | ■■ | ■■ | ■■ | ■■ |
| Component validation | ■■ | ■■ | ■■ | ■■ | ■■ | ■■ | ■■ | ■■ | ■■ | ■■ | ■■ | ■■ | ■■ |
| Tool availability | □□ | □□ | ■■ | ■□ | ■■ | ■■ | ■□ | ■■ | ■■ | □□ | □□ | ■■ | ■□ |
| Information sharing | ■■ | ■■ | ■■ | ■■ | ■■ | ■■ | ■□ | ■■ | ■□ | ■■ | ■■ | ■□ | ■□ |
| Mentioned = ■■ | | | | | Partially mentioned = ■□ | | | | Not mentioned = □□ | | | | |

The architectural representation of the proposed model is shown in Fig. 1. The Proposed Model is explained in detail with valid Component selection along with its algorithm and prototype based on variability management using reusability and user feedback. Therefore, the proposed model bridges gaps from the user requirements identification and validation in a system based on reuse and restricting with complete documentation to manage variability. This helps in managing the complexity and resources of HCS during developing a series of HCS products from requirements to validation.

Thus, the proposed model is composed of two processes as in any other SPLE process *i.e.*, domain engineering and application engineering. Domain engineering controls the development and maintenance of the domain and its related product development aspects like designs, features, and variability management. Moreover, all the aspects of the domain are managed in this process. On the other hand, the application engineering process controls the application-related tasks and aspects. The analysis of the application strategies like business goals and marketing strategies is also considered. After that, the application designing, implementation, and testing of the software variant are done in this process. The main components in the proposed model include dependency evaluation, variation management, documentation, and traceability testing. The problems identified in the previous versions include the lack of documentation for the component's selection and test suitcases pickups along with the end-user requirements. These requirements help the developers to provide the software of desired quality by tracking the requirements back to ensure the existence of all the functional and non-functional requirements in the system.

Moreover, the proposed model provides the classification of identified variations and commonalities based on their dependencies. These dependencies provide the list of the

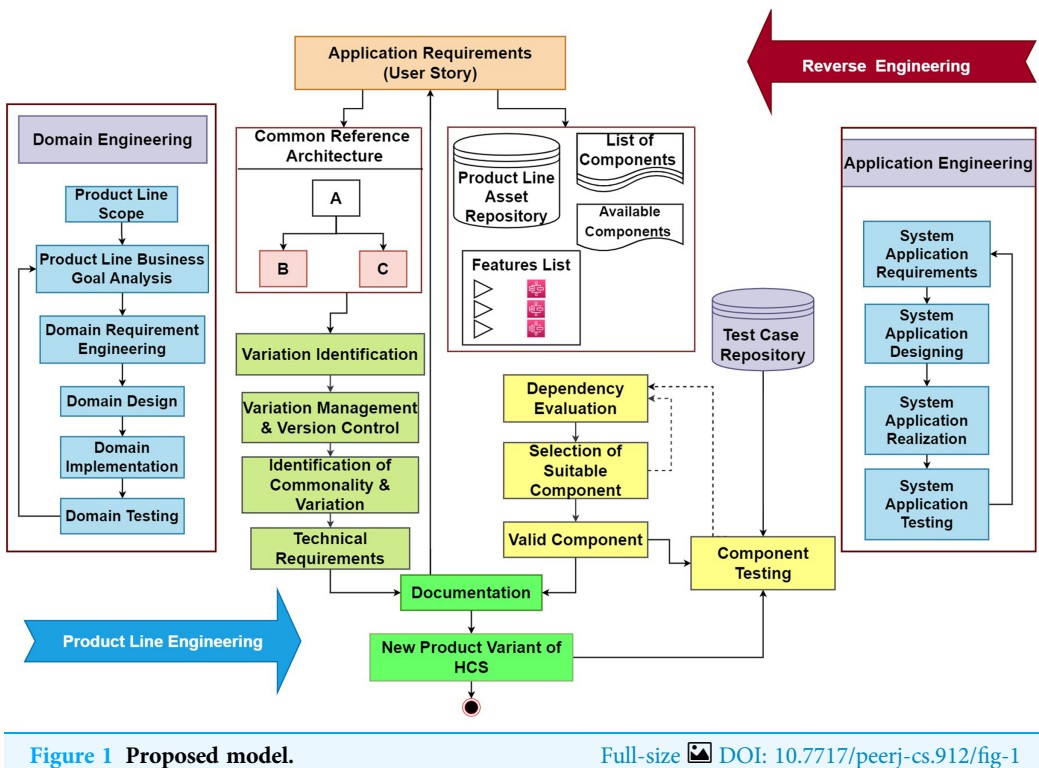

**Figure 1 Proposed model.**

dependent features for the selected component. A detailed discussion about the components of the proposed model is given below.

## Main entities of proposed process model

This section discusses the entities that are part of the proposed model. These entities are important to understand the complete working of the proposed mechanism. We used QeAPLE as a basic tool for component selection and validation. For task allocation, design, and development work synchronization as well as team coordination and communication and documentation version management, we have used a team server foundation repository with a prototype repository to align all the activities of the proposed model.

### Application requirements

When a new product or its variant is going to be developed, the very first thing is requirement gathering. These requirements are the instructions from the end-user or from the market that must be incorporated in the software going to be developed. For correctness and completeness, we consider the diverse perspective of stakeholders and involve stakeholders during requirements analysis and prioritization. Whenever the new requirements are gathered from the users, these requirements are checked in the domain assets repository based on cased based reasoning steps *i.e.*, to identify new requirements based on domain expert review and experience, to find similar requirements for reuse and restricting from a repository, modified requirements according to a new system and refine non-similar requirements to get complete and correct requirements. This improves the relevant selection of components for reuse and restructuring of

components with high productivity. After the selection of the components and features, the components are checked for their dependency. The component with the least dependency is selected from the list of identified components against each requirement.

### Common reference architecture

Any company offering or maintaining the SPLE process has a generic architecture that includes all the core functionalities. These functionalities or features are then tailored according to the requirements of the end-user to make a new variant of the existing domain. This will help the developers to tackle the new product more efficiently. The architecture is also used for the identification of commonalities and variations for the new product. These variations are done in the form of classes and stored in the documentation of that particular product.

### Variation and commonalities identification

When the requirements for the new product variants are received from the end-users, these requirements are then moved towards the generic domain architecture and product domain version control. From these modules, the variation and commonalities from the previous versions are identified. The identification for these variations is very important as these provide the identification face to the various versions of the product domain.

### Component selection

According to the received requirements, the components need to be selected from the database of the domain assets. These lists of components are then further sorted into single components list. These components have a list of features' information related to the product domain. These features are allowed to be reused in every variant corresponding to that product domain. We used steps of "case-based reasoning" which were adopted from the study (*Ali, Iqbal & Hafeez, 2018*; *Ali et al., 2021b*). The interfaces are of the QeAPLE prototype tool is depicted in Figs. 2 and 3. These interfaces of the prototype describe the functionalities of the component selection after identification of changes in HCS based SPL systems using case-based reasoning steps as explained earlier with the involvement of experts and stakeholders.

### Dependency evaluation

This is one of the major portions of the proposed process model in this research work. This module ensures and provides details about the dependency of the most suited component to the requirements with other selected components in the software. The main objective of this module is to clear the dependency of the most suited component or feature. This module finds the most suited component of the least dependency of the assets and then forwards the component to the next phase.

### Component testing

The selected components then undergo the testing phase before the integration of these components to form a final product. The tests are selected from the test suits, a big repository, for the retesting of the components. The main objective of this module is to

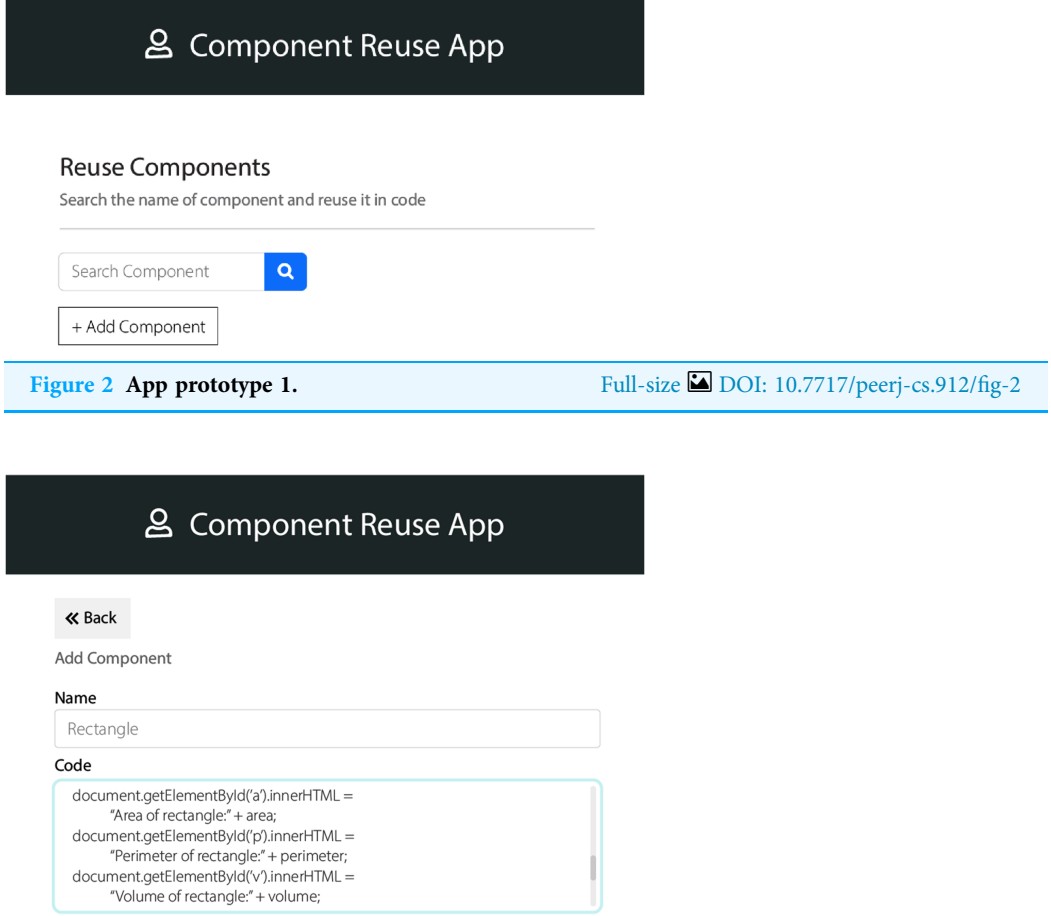

**Figure 2** App prototype 1.     

**Figure 3** App prototype 2.     

ensure the desired quality of the product. According to the requirement and component, the suitable test suit is extracted and applied to the component. If the component does not conform to the required functionalities, the component is then rejected otherwise it is selected for the integration.

### Test suit cases repository

This is another repository for the particular product domain. This repository is mainly composed of the test cases corresponding to the components of the product domain. These test cases are classified according to the level of non-functional requirements of the end-users and the type of functionality it offers. These test cases are selected on the go when a new component needs to be entered into the product. The interface is of the QeAPLE prototype tool is mentioned in Fig. 4.

### Documentation

This is the second most important module in the proposed process model. The documentation provides the facility to store the initial details of the new variant of the

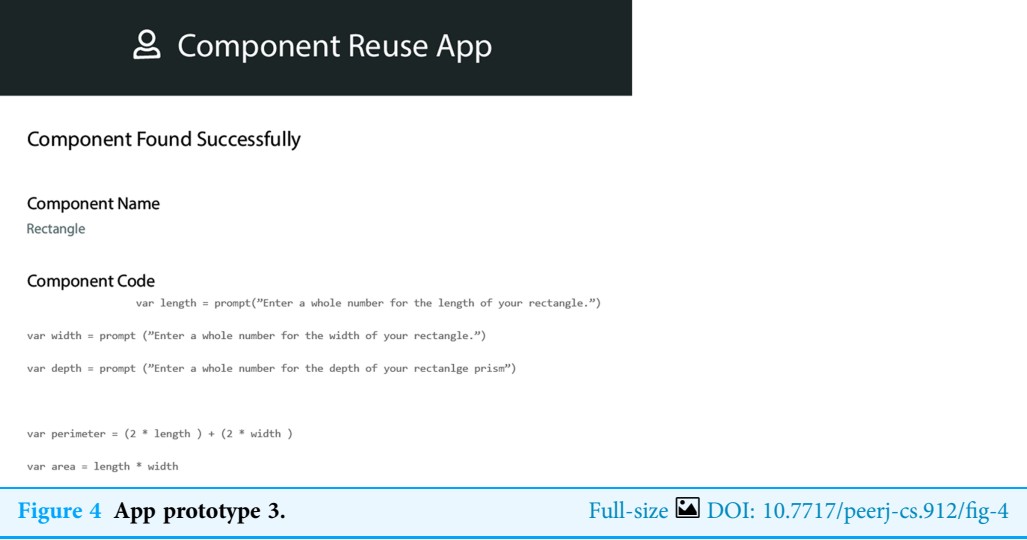

**Figure 4  App prototype 3.**           

product. Along with that it automatically includes the technical details about the products. Furthermore, this documentation helps to ensure the existence of all requirements in the product variant.

## Flow of the proposed process model

This section discusses the flow of the proposed model to elaborate on the beneficial outcomes of the proposed model. The complete state transition diagram of the proposed process model is shown in Fig. 1.

The process starts with gathering the requirements. Furthermore, to remove the ambiguity these requirements are made clear by using any of the best requirement gathering methods one of which is proposed by (*Geogy & Dharani, 2016*). After the collection of the requirements, these requirements are further provided to the generic domain architecture and the domain asset repository. The generic domain architecture provides the detailing of the functional and non-functional properties of the domain product, and this helps in the extraction of the design of the new variant going to be developed. Furthermore, it also helps the identification of the commonalities and variations for the new variant.

After the identification of variants, these variations are further moved to the variation management and version control module where the new version under the corresponding class is stored. After that, the requirements and the variations for the new product variant are added to the documentation maintained for that particular product version. This will help the developer to maintain the software and provide a valid update according to the market needs and requirements.

For the selection of the most suitable components and features that conform to the new requirements for the new product variant, the domain asset repository is used. In that repository, the most suitable components are filtered out among the lists of the components. After the selection of the most suitable components and features, the selected components are provided to the dependency checker module that confirms the

dependency of the selected module. This process continues in the iteration, and each component with the least dependency is finally selected at this stage.

After the selection of the least dependent components and features, the next step is the integration of these components to provide the desired software. But before the integration of these components, there is a phase where these selected components are get tested using the predefined test cases. These test cases are provided by the test case repository. This repository provides the test cases based on not only the functional properties of the product but also encounter the non-function aspect of the new product variant. Thus, it ensures the desired quality of the product. Afterward, the tested components are allowed to integrate while misfit or failed modules are again turned back and for them, replacement is arranged.

After the completion of the product, the used components and their corresponding test cases are stored in the documentation that is maintained for that particular product variant.

# EXPERIMENTAL EVALUATION

This section provides a discussion about the empirical evaluation of the proposed model. For that, an experiment is conducted in which the proposed approach is evaluated. The evaluation is made regarding the ease with which the proposed approach can understand and adapt by the practitioners, expected effort required to execute the proposed model, quality achievement of end-product achieved by using the proposed model, complexity reduced by the model for maintenance of end-product variant and improved version management for variants. The experimental details, conducted for the validation of the proposed approach are discussed below.

## Experiment design

The main objective of this evaluation is to know how it affects the development process of SPL; the experiment is conducted to compare the proposed model with one that is closely related to our approach (*Mollahoseini Ardakani, Hashemi & Razzazi, 2018*). The reason to select a single model for comparison is that mostly followed and adopted by researchers and industrialists respectively. And have lacked some of the main features in the selected model relevant HCS variability management by mapping requirements and validation activities after the identification of a relevant selection of components.

In this experiment, the proposed process model is used by the treatment group and the previously proposed process model *e.g.* (*Mollahoseini Ardakani, Hashemi & Razzazi, 2018*) by the control group. The comparison of both models will allow a better understanding of the improvement of the proposed model with the previous one. The selection of the previously proposed approach is based on the following reasons.

**Practical Relevance:** The process model proposed in *Mollahoseini Ardakani, Hashemi & Razzazi (2018)*, resembles the proposed process model in the sense that it also provides the integration of agile in the AE of the SPL process. The comparison will provide validations about the practitioners' aspect from adopting the proposed process model.

**Table 2 Null hypothesis.**

| RQs | Hypothesis |
| --- | --- |
| RQ1 | H0: There is no difference between the existing and proposed model with respect to ease of adaptability and understandability. |
| RQ2 | H0: There is no difference between the two models based on the required effort to execute various phases of model. |
| RQ3 | H0: There is no difference between the existing and proposed models with respect to the achievement of desired quality product variant. |
| RQ4 | H0: There is no difference between the two models corresponding to the maintenance complexity. |
| RQ5 | H0: There is no difference between the proposed and the existing models based on the improvement in the version management of the product variant. |

**Time Limitation:** There are some other SPLE based frameworks and models, but due to the shortage of time, this research work is confined to the comparison with only one proposed work.

**The Goal, Research Questions, and Hypotheses:** The goal of this experiment is the comparison of a proposed process model with one of the existing process models (*Mollahoseini Ardakani, Hashemi & Razzazi, 2018*) based on the ease in understandability, required effort, desired quality achievement, required maintenance complexity, and improved version management matrices. Depending on these comparison scales, the following research questions are derived.

**RQ1:** Does the ease of adaption and understandability is improved?
**RQ2:** Does reducing the effort required to execute different phases is reduced?
**RQ3:** Does the development of desired quality product variant is achieved?
**RQ4:** Does the maintenance cost and effort of the developed product are minimized?
**RQ5:** Does the variation management of the product is increased?

The next step is the formulation of the hypotheses required to be approved or disapproved based on the experimental results. The null hypotheses of the experiment states that there is no difference between both proposed models based on the degree of ease, required effort, desired quality achievement, maintenance complexity, and improved version manageability. The definition of the null hypotheses for the defined research questions is given in Table 2.

### Independent and dependent variables

In any empirical experimentation, there are two types of variables definition *i.e.*, dependent variable and independent variable. The change is done in the dependent variable and its effect is measured in the independent variable. As the name suggests, the dependent variables are the variables that are dependent on treatment and show some behavior on getting change. The deviation of this change is measured on independent variables. In this experiment, the dependent variable is dependency evaluation while selecting the component, automatic initial documentation of user stories, traceability orientation testing of end-product, and dependency matrices-based version management of components. Independent components in these experiments are ease of adaptability and understanding,

**Table 3 Independent variables.**

| NO# | Independent variables |
|---|---|
| 1 | Ease of adaptability and understanding |
| 2 | Required effort |
| 3 | Ability to achieve desired quality product variant |
| 4 | Maintenance complexity |
| 5 | Version management of the product variant |

**Table 4 Dependent variables.**

| NO# | Dependent variables |
|---|---|
| 1 | Dependency evaluation while selecting the component |
| 2 | Automatic initial documentation of user stories |
| 3 | Traceability orientation testing of end-product |
| 4 | Dependency matrices-based version management of components |

required effort, ability to achieve desired quality product variant, maintenance complexity, and version management of the product variants. The selected dependent and independent variables are shown in Tables 3 and 4 respectively.

### Experiment case

A case is a contemporary phenomenon for a better explanation in its real-life context (*Geogy & Dharani, 2016*). In this research work, a case is a course project conducted at COMSATS University Islamabad, Pakistan with two groups of students. These are the students who have studied the courses including the knowledge of coding, architecture, agile methodologies, and have some knowledge about the product line engineering processes and HCS. To remove the biasness, these students were all provided with definite classes in SPL and a HCS. Each group is composed of 30 students. The group of the first 30 students is named group A and the group of other 30 students is named group B. Group A is a control group while group B is the treatment group. A control group is a group that is used to measure the effect of change when the newly proposed approach is applied to the treatment group. Group A apply existing method on the given requirements of projects for new HCS product development based on APLE with complete previous version information. Similarly, group B developed product based on the steps of proposed model. All the participants were trained according to their methods which they apply during the development of HCS for a high-quality product. After the training of all the students, they applied their methods based on APLE on HCS development. Further, 15 subgroups were formed in each group *i.e.*, 2 students per group. Each group was given the same domain line project idea of developing and maintaining the inventory system product line. Group A followed the existing process model to manage the domain and to generate a new variant. While group B was given the proposed process model to develop and maintain the product line and its corresponding variant.

Summarizing the above discussion, the case is an activity that is performed in this experiment to check the worth of the proposed process model based on the matrices selected as the independent variables mentioned below:

- Ease of adaptability and understanding
- Required effort
- Ability to achieve desired quality product variant
- Maintenance complexity
- Version management of the product variants

### Experimental process

The main steps of the experiment are described in Fig. 5. The first step is to provide the students and team of selected organization project requirements are collected and transferred to every member of the company using various tools like Microsoft Teams, Cooja, *etc.*, for the basic details about the tasks they must perform. The reason to adopt various methods for communication used instead of single platforms is that the team and students participating in the experiment were distributed location-wise and have different communication languages and use a different medium for communications. After providing them with the required knowledge, the total number of 60 students was divided into two groups labeled (30 in each group) as *i.e.* A (treatment group) and B (control group). The next step after the division of the group is the provision of the details about the existing SPLE process and model to the control group and the proposed process model to the treatment group. After all the initial setup and provision of details, students are allowed to develop and maintain an inventory management system as a domain product and to allow the extraction of the various product variants. The domain development and maintenance are lengthy tasks. So, to provide the students with ease, an already developed domain product was taken as a test-bed. This domain product line is provided by a software company named Alachisoft located in Islamabad. After that, each group was asked to provide a new product variant from the domain assets using both models.

### Participants

There are some constraints during the selection of the participants for the software experiment. It is difficult to receive relevant outcomes if the experiment has insufficient participants and if the sample is not representative enough, then test effects can be debated. *Ro & Kubickova (2013)* suggest that in various disciplines students are used as an experimental subject and lots of debates are taking place for many years among the scientific community of using the student as a research subject. It is an extended debate in the research network for treating students as subjects in case studies and experiments. Participants selected for the execution of the experiment were third-year students who have studied agile development methods, software engineering, and software architecture. Along with that these students also have special courses for the knowledge of SPLE and HCS. The required tasks for the execution of the experiment are provided to the students in the fall semester from Sept 16, 2019, to Nov 28, 2019.

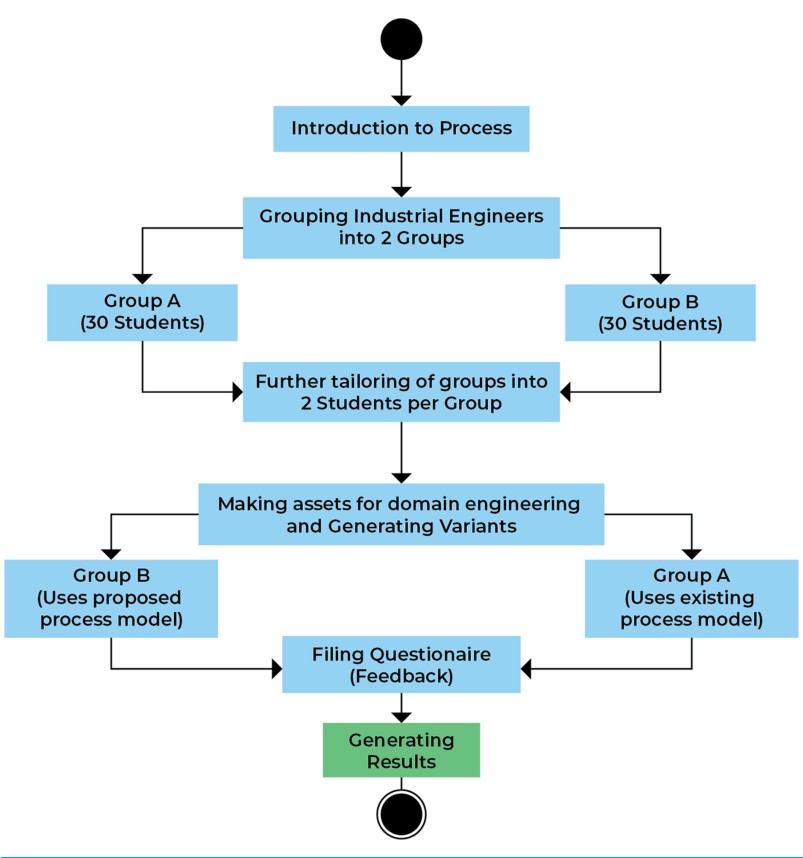

**Figure 5 Experimental process.**

To remove the biasness, the selection of the students was made randomly, and it is ensured that all the students have the approximately same skill set. According to the setup, the control group experimented, using the existing process model, and the treatment group experimented using the proposed process model. For the evaluation of the skill level and experience of the students selected as participants, a questioner was used. Most of the participants undergo their BS final projects. Among 60, 32 students were involved in industrial projects, 18 students performed excellence in their bachelor's degree and were awarded medals. Furthermore, these students were also asked if any of them has undergone any open-source project. In which six students admitted that they have performed open-source projects. Finally, the students were asked to mention their level of expertise between beginner, mediator, and experience in software engineering. Among them, 26 students went with beginners, 22 students said they are a mediator, and the remaining 12 students go with experienced. Student demographic information is shown in Table 5.

### Algorithm

The purpose of the algorithm is to identify the parameters like selecting suitable components. This algorithm helps practitioners in the selection of less dependent components. Developed a tool as a prototype for the QeAPLE in which this algorithm is implemented. Requirements are the input for the algorithm and the list of the least

**Table 5 Student demographics information.**

| Courses | Academic projects | Industry projects | Open-source projects | Experience level |
|---|---|---|---|---|
| ASE, SDLC, HCS, SPLE | Less than 3 (28 Students) | No project (19 Students) | No projects (37 Students) | Expert (5 Students) |
| ASE, SDLC, HCS, SPLE | More than 3 or less than 8 (22 Student) | Between 1 and 5 (35 Students) | One to five (21 Students) | Mediate (40 Students) |
| ASE, SDLC, HCS, SPLE | More than 8 (10 Students) | More than 5 (6 Students) | More than 5 (2 Students) | Beginner (15 Students) |

---

**Algorithm 1 Selection of suitable components.**

**Input:** RQS  List of Requirements

**Output:** $M_{LD}$ List of Least Dependent Module

1. RQS: $\{R_1, R_2, R_3, \ldots, R_n\}$
2. Modules: $\{M_1, M_2, M_3, \ldots, M_m\}$
3. Modular_Dep ← N  //Assign Dependency Value
4. $M_{Suit}$ ← Ø
5. $M_{Sel}$ ← Ø
6. $M_{LD}$ ← Ø
7. **For each** r ∈ RQS
8.     **For each** m ∈ Modules
9.         **if** (r ⊆ m)
10.             **then** $M_{Suit}$ ← $M_{Suit}$ ⋃ m
11.         **End For**
12.     **End For**
13. **For each** s ∈ $M_{Suit}$
14.     **if** (s ≤ Modular_Dep)
15.         **then** $M_{Sel}$ ← $M_{Sel}$ ⋃ s
16.     **End For**
17. **For each** x ∈ $M_{Sel}$
18.     **For each** y ∈ $M_{LD}$
19.         **if** x < y
20.             **then** $M_{LD}$ ← $M_{LD}$ ⋃ x
21.                 $M_{LD}$ ← y
22.         **End For**
23. **End For**
24. **Return $M_{LD}$**

---

dependent module is the output of the algorithm. At the initial stage, dependent variables are initialized to null values. The steps of the algorithm are mentioned below:

## Analysis of experimental data

This section contains a discussion about the statistical analysis of the data gathered from the experiment by filling questioner from students. The questioner helps in collecting and analyzing data after experimenting to evaluate the effectiveness of the proposed model and performance of participants of both groups using the proposed model and existing model. The effectiveness of the proposed model was used to analyze whether the identified

| Measure | Data normality | | Null hypothesis *P*-value | A12 | Cohens D |
|---|---|---|---|---|---|
| | Group A (*P*-value) | Group B (*P*-value) | | | |
| Easy to understand | 0.00032 | 0.0019 | 0.01 | 0.64 | 0.51 |
| Effort required | 0.0005 | 3.931e−05 | 0.03858 | 0.62 | 0.47 |
| Better quality achievement | 5.095e−05 | 0.00037 | 0.00681 | 0.67 | 0.67 |
| Maintain complexity | 0.000667 | 0.00335 | 0.031 | 0.64 | 0.51 |
| Improved version management | 0.00049 | 0.00093 | 0.03803 | 0.625 | 0.46 |

**Table 6 P value.**

Notes:
   A12: In comparison between Group A (Control Group) and Group B (Treatment Group) where *P*-value < 0.5, If A12 < 0.5 then Group A is better than Group B else if A12 > 0.5 than Group B is better than Group A.
   Cohens D (d): If *d* >= 0.8 than significance is large, if *d* <= 0.5 than significance is medium and if *d* < 0.2 than significance will be small.

problems from the literature were resolved. Similarly, the performance of participants helps in proofing satisfaction level of the participants in terms of understandability, effort, time, and cost. To evaluate the results, a quantitative analysis procedure is adopted. The analysis of the data starts with the data normality check. For this purpose, several empirical tests including qqnorm, qqline, Shapiro wilk, and Anderson darling test are executed. The p-value obtained from the tests is shown in Table 6. As the p-value is less than the significance level, which shows that the data is not normal. So, to validate such data, the Mann–Whitney *U* test is executed for the comparison of the independent variables (*Ghasemi & Zahediasl, 2012*).

### RQ1: easy to adapt and understand

The experimental data obtained for easy understandability and adaptability is normally distributed as shown in Table 6. Therefore, to test the hypothesis formulated for RQ1, the Mann–Whitney *U* test is applied, and to find the direction of change, the A12 test is applied (*Narasimhan et al., 1986*). The results of these tests are clearly described in Table 6. As shown in Table 6, the *p*-value for group A and group B are 0.00032 and 0.0019 for the variable easy to understand. Furthermore, the graphical representation of these results is shown in Fig. 6.

   According to the results of the test, there is a significant difference between the existing and the proposed process model based on the ease of understandability and adaptability. This shows the superiority of the proposed process model over the previous one. Along with A12, the mean values were also calculated by filling the questioner from the subjects, which also supports the arguments about the excellence of the proposed process model. Finally, the null hypothesis formulated for RQ1 is rejected and as a result, the alternative hypothesis is accepted.

### RQ2: expected effort

To calculate the effort required to follow the process model, the total time consumed for executing the proposed model is selected as a parameter. The total time required to follow for each activity is calculated and then added to get the overall time. After the execution of the experiment, the subjects are asked to fill the questioner to get their

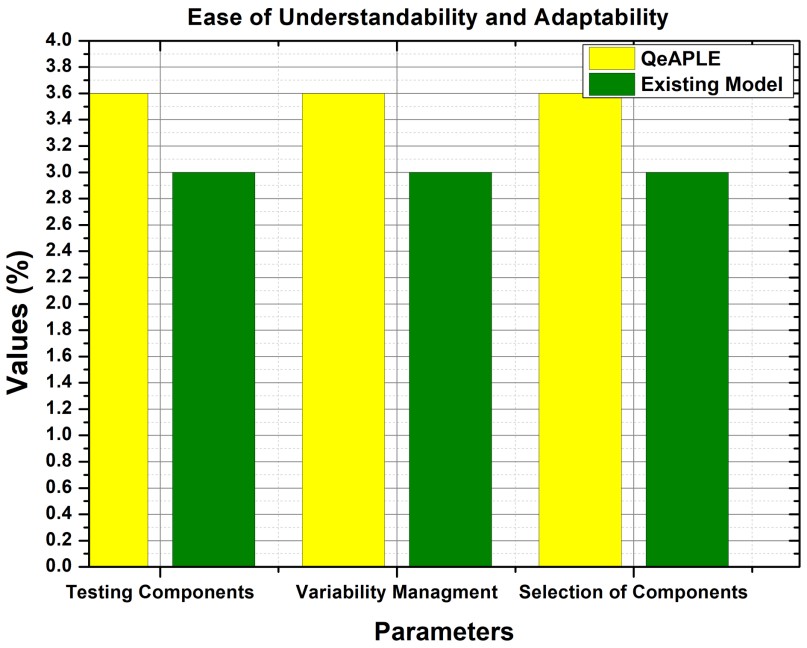

**Figure 6** Ease of understandability and adaptability.

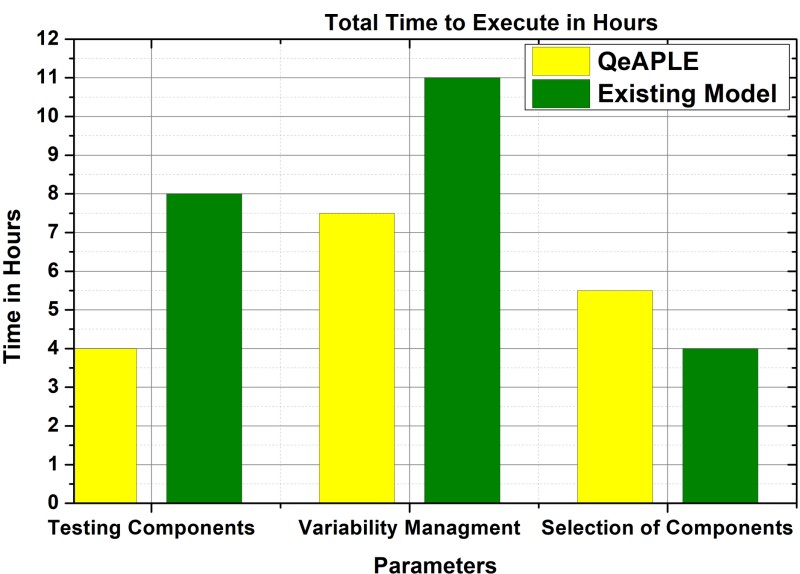

**Figure 7** Expected effort.

opinions. After getting the responses from the subjects, the normality test is applied to it which finds out that the data is not normally distributed. To evaluate the proposed hypothesis for RQ2, the non-parametric test *i.e.*, Mann–Whitney *U*, is applied to the data.

The result obtained from the statistical tests is shown in Fig. 7 and describes the time required to complete different tasks. To find the direction of the significance for both the process models, the A12 test is applied, the result of which is shown in Table 6. To find the magnitude of the difference the Cohens-D test is applied, the result of which is shown

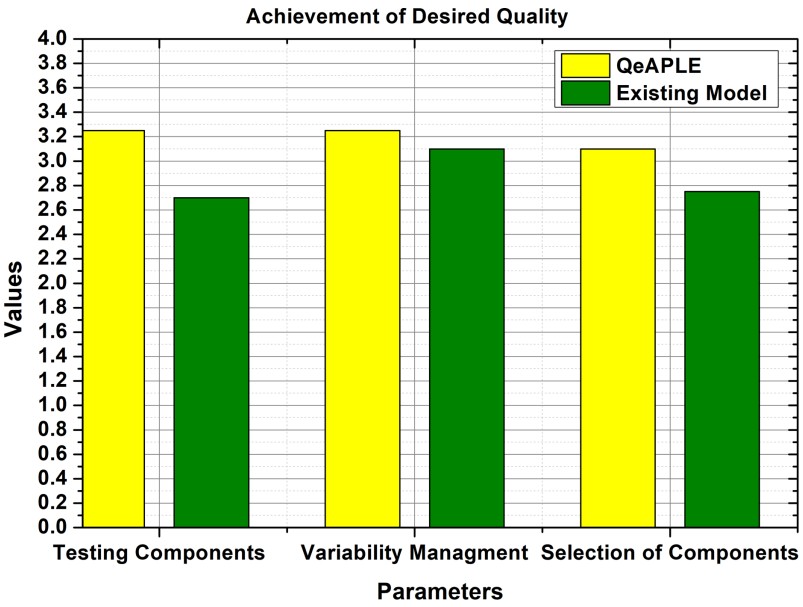

**Figure 8 Achievement of desired quality.**

in Table 6. The test results of Cohens-D show that there is a medium difference between both the process models. Finally, the results of the experiments reject the null hypothesis and thus the alternative hypothesis is accepted.

### RQ3: Better quality achievement

To calculate the degree to which the quality of the product variant is achieved for both the process model, the specifications of the parameters were collected and shown to the practitioners, practitioners filled the questioner after reviewing the requirements for the product and the new product variant. To check the normality of the data, the normality test was applied which provides the details about the normality of the data. To evaluate the hypothesis proposed for the RQ3, the non-parametric test was applied to the data whose p-value is shown in Table 6. The result obtained from the statistical tests is shown in Fig. 8.

Furthermore, to find the direction of the significance, the A12 test is applied which shows that the proposed process model is more effective and good as compared to the existing one. After finding the direction, the next check was the evaluation of the magnitude of the difference between both the process models. For this purpose, the Cohens-D test was applied, which proves that there is a medium difference between both the models. Therefore, the null hypothesis is straight-away rejected, and the alternative hypothesis is accepted.

### RQ4: Maintenance complexity

To evaluate the total amount of complexity for the maintenance and updating of the product, every group was asked to make some changes in the newly developed product variant. Here they first need to identify the corresponding change, then selection of the proper component, and finally the testing and integration. The evaluation parameter

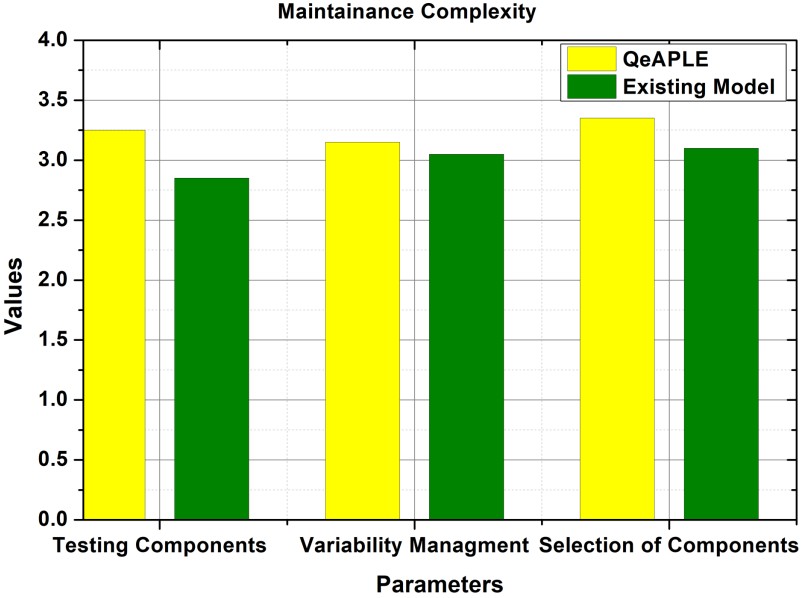

**Figure 9  Maintainance complexity.**    

selected for the validation of maintenance complexity was the total time, taken by the groups to maintain or incorporate updates in the newly developed product variant. To get the statistical data, the questionnaire was filled by the subjects, and the total time taken for the incorporation of updates was recorded as shown in Fig. 9. The incorporation of practitioner's advice is important here to acknowledge the accuracy with which the updates are performed in the developed system. The mean values gathered from the test undergoes for the normality test. The normality test provides the information that the data is not normally distributed and thus for the evaluation of the hypothesis, the non-parametric test will be used.

After checking the normality of the data, the Mann–Whitney $U$ test was applied whose result is shown in Table 6. This shows that there is a difference between both approaches as the $p$-value is less than 0.5. To find the direction of the magnitude, the A12 test is applied which shows that the proposed process model is better than the existing model. Further to check the significance of the difference, the Cohens-D test is applied which shows that there is a medium difference between both the process models. Based on the analysis, the null hypothesis proposed for the RQ4 is rejected and the alternative hypothesis is accepted.

The values obtained from the experiment were then checked for normality. The normality test shows that the experimental data is not normally distributed. To check and validate the hypothesis the non-parametric test $i.e.$, Mann Whitney $U$ test is performed on the experimental data. The result of this data is shown in Table 6. As the results describe the value of $p$-value is lower than 0.5, which means there is a difference between both the process models. To check the direction of the magnitude of change, the A12 test is applied. A12 shows that the proposed process model is better than the existing process

model. To check the significance of the difference, the Cohens-D test is applied which shows that there is a medium difference between the two-process model.

Based on these findings, the null hypothesis proposed for RQ5 is rejected and as a result, the alternative hypothesis is accepted.

All the experiment is based on the questionnaire which is attached in Appendix A. For the reliability of the questionnaire, we performed reliability statistical analysis using the SPS tool by Appling reliability test to check data biasness and accuracy. For the reliability test, we use SPSS 23 tool and automatically extract the results. The participants' information and the result of the statistical test are in Table 6.

## Threats to validity

This section aims to discuss the threats to the validity of the experiment performed according to guidelines provided in *Heck & Zaidman (2018)*, *Lindohf et al. (2021)* and *Kiani et al. (2021)*.

### Construct validity

The main focus of this threat is the ability to measure the required facility operationally without error. In this experiment, the main objective is to measure the efficiency of both process models. Therefore, the same evaluation factors are defined for both models. Furthermore, the subjects are clarified that this activity will not perform any role in the grading of any subject. So that it would not cause any biasness. To make the experimental hypotheses private, the information about the experimental hypotheses is kept hidden from the subject to avoid any type of biasness with the researcher. Hence, to avoid error and biasness during experiment while using both methods. The participants of the proposed model and existing model were fully trained before the execution of methods during development of HCS.

### Internal validity

The main aim of this threat is the problem of biasness caused by the casual relationship between the experiment subject and the researcher. To make a clear evaluation of the proposed model, the experiment is done very carefully by providing all the necessary tutorials and labs to the experiment subject. Furthermore, to overcome the biasness, complete random groups were designed and further the students were advised to actively participate without being afraid of any grade manipulation. To ensure the complete presence of the students they are also asked to further provide their values and opinion about how the process can be improved further. The participants performance was not influenced with any type of relations and participants of both groups separately performed development activities without knowing each other's in different times and environments.

### External validity

The main concern of this threat is the generalization of the results concluded from the experiment. The experiment was conducted using the students belonging to COMSATS

University. Therefore, the participants used for the execution of the experiment are not professionals. The reason behind the selection of students as an experimental subject lies in the least availability of professionals from the industry. Furthermore, most of the empirical research in software engineering uses student and experimental subjects for the execution of the experiment. Finally, the nature of the experiment doesn't require the professional to be part of the experiment.

### Conclusion validity

Violating the statistical test assumption may result in a conclusion not much accurate. The experimental data is on an interval scale that could be a risk for statistical tests for the achievement of better results. The non-parametric Mann–Whitney $U$ test is used for making these assumptions. Our sample size fulfills the criteria for the statistical test but is not too large because of large sample size increases the power of the test.

## CONCLUSION AND FUTURE WORK

Many software development process models are described in the literature that tends to join the SPL and APL to provide the comprehensive end product variant in large industries. These process models lack the proper documentation, not ensuring the quality of the components and details about the selection of the features based on the required specification. To address these problems, a hybrid APL model, QeAPLE is proposed that provides support for HCS by evaluating the dependency of features before making the final selection. It provides a comprehensive way for the selection of the components that are least dependent upon each other. Moreover, it also provides well-detailed documentation along with the testing of the selected components to clinch the quality of software and sparing time of the post-testation of the released product variant.

The main augmentation of this research effort comprises of:

- The presentation of innovatory knowledge about the agile, SPL, and their integration for the development of systems especially for HCS systems.
- The proposition of the new hybrid process model allows the incorporation of SPL and agile processes together with the development support for HCS using the least dependent component selection.
- The evaluation of the proposed approach using the use case study and practitioner close-ended interviews along with the empirical evaluation executed using students as subjects.

The possible future works could be:

- The main future direction could be the shortness of the time taken for the selection of the components.
- Could be the introduction of AI technology result in better selection of component that is least dependent and highly effective for the required requirements of a variant.

### Funding
The work reported in this manuscript was supported by the National Natural Science Foundation of China under Grant 61672080. The funders had no role in study design, data collection and analysis, decision to publish, or preparation of the manuscript.

### Grant Disclosures
The following grant information was disclosed by the authors:
National Natural Science Foundation of China: 61672080.

### Competing Interests
The authors declare that they have no competing interests.

### Author Contributions
- Tehseen Abbasi conceived and designed the experiments, performed the experiments, analyzed the data, performed the computation work, prepared figures and/or tables, and approved the final draft.
- Yaser Hafeez conceived and designed the experiments, performed the experiments, analyzed the data, performed the computation work, prepared figures and/or tables, authored or reviewed drafts of the paper, and approved the final draft.
- Sohail Asghar conceived and designed the experiments, analyzed the data, prepared figures and/or tables, authored or reviewed drafts of the paper, and approved the final draft.
- Shariq Hussain analyzed the data, authored or reviewed drafts of the paper, and approved the final draft.
- Shunkun Yang performed the experiments, authored or reviewed drafts of the paper, and approved the final draft.
- Sadia Ali performed the experiments, prepared figures and/or tables, and approved the final draft.

### Data Availability
The code, developed in Mango schema (as component library) and library in Node.js, used for this study, is available in the Supplemental File.

### Supplemental Information
Supplemental information for this article can be found online at http://dx.doi.org/10.7717/peerj-cs.912#supplemental-information.

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
