# Peer review of "Towards a component-based system model to improve the quality of highly configurable systems"

_PeerJ Computer Science, doi:10.7717/peerj-cs.912_

## Round 0.1 · original submission · Major Revisions

The authors are advised to carefully address all the comments and concerns mentioned by all reviewers with their revised version.

Reviewer 1 ·

Basic reporting

Needs thororugh proof read

Experimental design

OK

Validity of the findings

Can be enhanced further. The novelty and impact requires elaboration.

Additional comments

The work Towards a component-based system model to
improve the quality of highly configurable systems
is the current work and better presented in most of the aspects.
However, authors can further improve their work in the following sections,
1. The abstract can be improved with the significance of the results.
2. The introduction requires modification such as being concise.
3. Further authors may elaborate on the research contribution in the introduction.
4. Provided literature is better but authors may add a bit more on the literature.
5. The proposed framework is not readable easily.
6. Authors may elaborate about the data set they used.
7. A few of the references need attention to complete the required information.
8. Few sentences need careful attention, they are required to be rephrased such as " Hence, results show that the easy of ease of understanding and adaptability, required effort, high-quality achievement,
and version management is significantly improved i.e., more the 50 per cent as compared
to exiting method i.e., less than 50 per cent"
9. Please carefully check the sentence structure " da Silva et al. [35] presented a
185 new ability-based approach for scoping the product line details".
10. Few references are not relevant like 39, 40.
11. Authors please carefully use words Hypotheses and Hypothesis.

·

Basic reporting

The paper presents a Component-Based System Model to Improve the Quality of Highly Configurable Systems to support IT-related organizations. The paper is very well written and interesting concept. Following suggestions would be incorporated to Improve the quality of paper.


1. There are few unusual long statements that may exceed three lines. A thoroughly English language revision is a must, where many English language, spelling errors, grammar and punctuation errors are found.

2. The Abstract should clarify the evaluation criteria of the Component-Based System Model and the main findings and results of the evaluation quantitatively.
3. The suggested title should be clear and enlightening, and should reflect the aim and approach of the study.
4. A section should be added between the introduction and proposed model sections to clarify and emphasize the main contributions and its scientific justification and applicability, in this study.
5. There are some grammatical and typo mistakes of English in paper and need some minor adjustments.
6. There are some formatting issues in caption of figures and tables, please address them.
7. Enhance the pictures’ size for better presentation and readability.

Experimental design

satisfactory

Validity of the findings

Done in a better way

Additional comments

The “Research Problem” section must be under the heading of Introduction for better understanding of the proposed research.
“Literature Study” section is not exhaustive to narrate existing work in the identified domain so it is better to include 2 or 3 more paragraphs including their references.
Include more references focusing on highly configurable systems through software product line.

Reviewer 3 ·

Basic reporting

The author work is good. The abstract covers the complete paper. Author provided detail background of the work. Author also focused on the related literature. Author produced required figures and tables which are necessary for the explanation of the work.

Experimental design

The author work is related to the scope of the journal. Author introduced valid research gaps and later on the findings are adding to the domain of knowledge. Author design methodology in detail and each component is well explained.

Validity of the findings

The data and processes claimed by the author are valid and according to domain of the work. The result is verified statistically using different statistical tools.

Additional comments

The author concluded work in detail. The conclusion also providing readers a thinking about future work in this research direction.
Some comments given below which should be addressed in the final submission.

1) Correct spelling exiting into existing Line 28 and line 30
2) Author should adopt one method, i.e. either use abbreviation and full text for all or only use abbreviation except first time. For example, SPLE, HCS, QeAPLE etc.
3) Complete sentence at line 90. State-of-the-art ….. what next?
4) Author should read full paper for required English grammar corrections.

---

## Round 0.2 · accepted · Accept

We appreciate your work considering the comments and concerns of all reviewers.

Reviewer 1 ·

Basic reporting

OK

Experimental design

OK

Validity of the findings

OK

Additional comments

Authors have incorporated the suggested changes.

Reviewer 3 ·

Basic reporting

Author did good effort in updating paper according to comments.
Author also highlighted changes in the paper through submitting track changes file.
Author's response to comments in detail.

Experimental design

The experimental design is satisfactory.
The flaws are removed in revised version.

Validity of the findings

The focus is on real time problem solving. This is a validity of the finding by the author.

Additional comments

The paper is in good shape after required changes.